# Development of A Nanostructured Lipid Carrier-Based Drug Delivery Strategy for Apigenin: Experimental Design Based on CCD-RSM and Evaluation against NSCLC In Vitro

**DOI:** 10.3390/molecules28186668

**Published:** 2023-09-17

**Authors:** Xiaoxue Wang, Jinli Liu, Yufei Ma, Xinyu Cui, Cong Chen, Guowei Zhu, Yue Sun, Lei Tong

**Affiliations:** 1Department of Pharmacy, Mudanjiang Medical University, Mudanjiang 157000, China; xiaoxuew1997@163.com (X.W.); mayufei693@126.com (Y.M.); c1449320686@163.com (C.C.); z2328602734@163.com (G.Z.); sunyue02282021@163.com (Y.S.); 2The Affiliated Hongqi Hospital, Mudanjiang Medical University, Mudanjiang 157000, China; liujinli1986123@163.com; 3Department of Public Health, Mudanjiang Medical University, Mudanjiang 157000, China; xinyucui@outlook.com

**Keywords:** apigenin, nanostructured lipid carrier, NSCLC, NCI-H1299

## Abstract

Non-small-cell lung cancer (NSCLC) is the main cause of cancer-related deaths worldwide, with a low five-year survival rate, posing a serious threat to human health. In recent years, the delivery of antitumor drugs using a nanostructured lipid carrier (NLC) has become a subject of research. This study aimed to develop an apigenin (AP)-loaded nanostructured lipid carrier (AP-NLC) by melt sonication using glyceryl monostearate (GMS), glyceryl triacetate, and poloxamer 188. The optimal prescription of AP-NLC was screened by central composite design response surface methodology (CCD-RSM) based on a single-factor experiment using encapsulation efficiency (EE%) and drug loading (DL%) as response values and then evaluated for its antitumor effects on NCI-H1299 cells. A series of characterization analyses of AP-NLC prepared according to the optimal prescription were carried out using transmission electron microscopy (TEM), differential scanning calorimetry (DSC), X-ray diffraction (XRD), and Fourier transform infrared spectroscopy (FT-IR). Subsequent screening of the lyophilization protectants revealed that mannitol could better maintain the lyophilization effect. The in vitro hemolysis assay of this formulation indicated that it may be safe for intravenous injection. Moreover, AP-NLC presented a greater ability to inhibit the proliferation, migration, and invasion of NCI-H1299 cells compared to AP. Our results suggest that AP-NLC is a safe and effective nano-delivery vehicle that may have beneficial potential in the treatment of NSCLC.

## 1. Introduction

Lung cancer is a prevalent type of cancer throughout the world. According to the pathological type, it can be divided into two categories: non-small-cell lung cancer (NSCLC) and small-cell lung cancer. NSCLC accounts for 80–85% of total lung cancers, and with a low five-year survival rate, it is considered to be a major culprit in cancer-related deaths in the world’s population [1,2,3]. Treatments for NSCLC in its early stages include surgery, chemotherapy, and radiotherapy, but there are some potentially harmful side effects of chemotherapy and radiotherapy [4]. Natural active substances isolated from plants have been widely studied and focused on for their potent antitumor activity and low toxicity [5].

Natural compounds can participate in regulating the metabolic process of cancer cells through multiple targets and signal pathways to play an antitumor role [6]. AP, chemically known as 4′,5,7-trihydroxyflavone, is a naturally occurring polyphenol found primarily as a glycoside in chamomile flowers, parsley, bergamot, celery, artichoke, and kumquat [7]. Numerous studies have demonstrated the great biological potential and rich pharmacological effects of AP, including its anti-inflammatory [8], antioxidant [9], antibacterial [10], and antiviral [11] effects, and its use as a treatment for Alzheimer’s disease [12] and depression [13]. On top of that, AP is a remarkable bioactive substance in the treatment of NSCLC, as demonstrated by its ability to inhibit cell proliferation by suppressing promoter expression, inhibit angiogenesis and tumor growth by acting as a HIF-1α inhibitor, reduce metastasis, and induce apoptosis in NSCLC [14,15,16,17]. Nonetheless, it is greatly limited in clinical application due to its low aqueous solubility [18,19]. Moreover, the structure of the multi-hydroxyl group is unstable. Therefore, the key to solving the above questions is to find an effective drug delivery system.

In the early 1990s, solid lipid nanoparticles (SLNs) were developed as a new preparation based on nanoparticles [20]. The main components of SLNs are usually solid lipids, emulsifiers, and water, with particle sizes ranging from 50 to 1000 nm [21]. Limitations include low drug-loading capacity due to the crystalline nature of the lipids and the formation of a perfect crystal lattice. To remedy these deficiencies, a second generation of lipid nanoparticles, NLCs, was developed by adding liquid lipids to the formulation of SLNs. The regular arrangement of the perfect crystalline lattice of solid lipids in SLNs would be distorted by liquid lipids, generating more crystalline lattice defects and making it easier for the drug to enter the lipid skeleton, as well as reducing particle sizes, enhancing the drug-loading capacity, and decreasing the efflux of encapsulated substances during storage [22,23].

The ratio of each component in the prescription is crucial for achieving maximum efficacy of the preparation. Therefore, prescriptions with different ratios need to be thoroughly investigated and screened. Encapsulation efficiency (EE%) and drug loading (DL%) are usually important indicators for evaluating the quality and process of nanoformulations. EE% is the ratio of the encapsulated drug to the total amount of the drug, and DL% is the ratio of the encapsulated drug to the total amount of the preparation. Design of experiments (DoE) is a mathematical statistical method for arranging experiments and analyzing experimental data. It can be used to determine the maximum EE% and DL%, which is of extraordinary significance in maximizing the efficacy of AP. DoE can be implemented via factorial design and response surface methodology, such as 3^k^ factorial design, Box–Behnken design (BBD), and central composite design response surface methodology (CCD-RSM). When a large number of variables lead to a significant number of experiments, 3^k^ factorial design becomes ineffective, and BBD may also fail to capture extreme cases [24]. However, in comparison to the aforementioned designs, CCD-RSM has the capability to accurately predict results without being affected by lost data. In addition, the application of CCD-RSM in response surface methodology can simultaneously determine the interaction between different variables that affect the response value [25]. Because the data obtained using CCD-RSM show excellent and reliable prediction ability, CCD-RSM has been widely used to develop and optimize new prescriptions [26].

The objective of the study was to prepare AP-NLC (Figure 1) and optimize it using CCD-RSM to obtain the optimal prescription and perform a series of characterization analyses, followed by a preliminary evaluation of the in vitro anti-NSCLC ability of AP-NLC.

## 2. Results and Discussion

### 2.1. Determination of AP Content

#### 2.1.1. Choice of Maximum Absorption Wavelength

The gradient concentrations of the AP–methanol solution all had a maximum absorption peak at 335 nm with a gentle peak shape and small calculation error (Figure 2), so the maximum absorption wavelength was set at 335 nm.

#### 2.1.2. Drawing of Working Curve

The working curve regression equation of AP is A = 0.0748 C + 0.0026 (R^2^ = 0.9997), and the linear relationship is good within the concentration range of 1–12 μg/mL.

### 2.2. Single-Factor Experiment

Figure 3 exemplifies the variation in EE% and DL% with changing levels of different experimental factors. By setting the ultrasonic power at 150 W, emulsifier dosage at 0.5%, and emulsification time at 10 min, undissolved lipids were floating on the liquid surface, probably due to incomplete ultrasonication or insufficient emulsification. Therefore, the above experimental levels were discarded. After dropping the above levels and changing the ultrasonic power and emulsification time, EE% and DL% were almost unchanged, indicating that the ultrasonic power and emulsification time were not important factors affecting EE% and DL%. The emulsifier dosage was at the rate of 2.5% and the formulation system foamed, suggesting that too much was added. The lipid–drug ratio was increased from 20 to 30; DL% decreased, but EE% first increased and then decreased. The fluctuation of EE% and DL% caused by changing the solid–liquid lipid ratio was coherent. Therefore, it was finally determined that the emulsifier dosage (1–2), lipid–drug ratio (20–30), and solid–liquid lipid ratio (3–8) were set as the three experimental design factors as well as the high and low levels for CCD-RSM.

### 2.3. Optimization of AP-NLC

#### 2.3.1. ANOVA of The Model

Design Expert 13.0 was used to analyze the test results in Table 1, and the regression equation between EE% (Y_1_) and DL% (Y_2_) of AP-NLC was obtained as follows: Y_1_ = 83.26 − 2.75 A − 0.9209 B + 6.86 C − 4.46 AB − 1.29 AC − 1.83 BC − 4.06 A^2^ − 3.89 B^2^ − 1.48 C^2^, Y_2_ = 3.23 − 0.0965 A − 0.7046 B + 0.2592 C − 0.2200 AB − 0.1200 AC − 0.1825 BC − 0.1806 A^2^ − 0.0198 B^2^ − 0.0816 C^2^ + 0.1071 A^2^B.

According to Table 2, primary items A and C, interactive items AB and AC, and secondary items A^2^, B^2^, and C^2^ have a very significant impact on EE% (*p* < 0.05), while other factors have no significant impact. With reference to the F-value in Table 2, the following is the order of factors affecting EE% of AP-NLC: solid–liquid lipid ratio > emulsifier dosage > lipid–drug ratio. The *p*-value of the model is less than 0.0001, which proves that the response surface model has reached a very significant level, and the lack of fit item (*p* = 0.0710 > 0.05) is not significant. The R^2^ of the model is 0.9669, which shows that the test model fits well with the actual test, and about 96.69% of the results from the actual test can be explained by the fitting model. The adjusted R^2^ after correction is 0.9372, which is close to R^2^, indicating that the model has sufficient accuracy and predictability. Therefore, this model can be used to analyze and predict the influence of various factors on EE% of AP-NLC. Similarly, according to Table 3, the impact of all factors on DL% was significant, except for secondary item B^2^.

#### 2.3.2. 3D Response Surface and Contour

The effects of emulsifier dosage, lipid–drug ratio, and solid–liquid lipid ratio on EE% and DL% of AP-NLC were analyzed according to a 3D response surface and contour map. When one of the three factors is fixed, the influence of the interaction between the other two factors on the response value can be represented by contour lines and response surfaces. The results are shown in Figure 4, where the impact of the interaction between factors on the response values is more visually reflected through response surface and contour plots. When the 3D response surface is steeper and the contour lines are denser, the influence of the factors on the response value is greater. The interaction between two factors is weaker when the shape of the contour lines is closer to a circle; conversely, the interaction is stronger when the shape is closer to an ellipse. In addition, the color of the 3D response surface and contour plot transitions from blue to red, which represents that the value of EE% or DL% changes from low to high, and the value in the red area is closer to 100%. The red dot indicates the maximum value that EE% and DL% can achieve under the current numerical limits of the three factors (A, B, C).

#### 2.3.3. Determination and Validation of The Optimal Prescription

According to the results of CCD-RSM screening, the best prescription was determined: the emulsifier dosage was 1.33%, the lipid–drug ratio was 20, and the solid–liquid lipid ratio was 9.52. The predicted EE% and DL% of this formulation were 90.13 and 4.40. The low EE% and DL% values may be attributed to the poor solubility of AP. Subsequently, three batches of samples were prepared; the relative deviation between the predicted value and the measured value was less than 5% (Table 4), denoting good predictive performance of the CCD-RSM for optimizing prescriptions.

### 2.4. Transmission Electron Microscopy

The AP-NLC prepared according to the optimal formulation was a yellowish suspension (Figure 5C). The nanoparticles were observed to be spherical or oval in shape by TEM with an average particle size of approximately 50 nm (Figure 5A,B).

### 2.5. DSC Analysis

DSC as a general tool can be used to probe the melting and recrystallization of drug crystals in nanoformulations [27]. Figure 6 represents the DSC curves of GMS, AP, their physical mixture, and AP-NLC lyophilized powder. Thermograms of AP show a melting peak around 366 °C. The melting process of GMS occurred at 80 °C and produced a strong melting peak. The above two melting peaks both appeared in the mixture of AP and GMS. The peak shapes of the DSC curves of AP-NLC and blank NLC were similar. However, the characteristic peak of AP was not detected in the thermograms of AP-NLC lyophilized powder, which indicated that AP did not exist in a crystalline state in the lipid matrix.

### 2.6. X-ray Diffractometry

XRD is a drug tool of pharmaceutical analysis used for investigating the structure of crystalline substances, and the peak positions of XRD can be exploited to determine the crystallinity of different samples [28]. Figure 7 is the XRD spectrum of AP, GMS, AP-NLC, blank NLC, and the physical mixture of AP and GMS. AP showed sharp characteristic peaks at 2θ angles of 6.98°, 11.19°, 14.18°, and 15.84°, verifying the strong crystallinity of AP, which is consistent with what has been reported [29]. The arrows indicate the locations of these diffraction peaks, which vary significantly among different samples within the red box. GMS showed two broad characteristic peaks at 2θ (diffraction angles) of 19.56° and 22.92°. The diffraction peaks of the physical mixture appeared to be a superposition of the characteristic peaks of AP and GMS. The XRD pattern of AP-NLC showed only the characteristic peaks of GMS and not the diffraction peaks of AP. The above results revealed that AP no longer exists in a crystalline state in AP-NLC and is almost amorphous.

### 2.7. FT-IR Analysis

Infrared spectroscopy can serve to study the interactions between nanoformulations and excipients that may occur during their preparation. The FT-IR spectra of AP, AP-NLC, GMS, and blank NLC are shown in Figure 8. The strong bands generated by the valence vibration of the hydroxyl group (-OH) at 3292 cm^−1^, the stretching vibrations of the carbonyl group (C=O) at 1653 cm^−1^ and 1608 cm^−1^, and the stretching vibration of the pyran ring (C-O-C) at 1355 cm^−1^ were classified as the characteristic absorption peaks of AP [30]. The absorption peaks of GMS at 3316 cm^−1^, 2914 cm^−1^, 2851 cm^−1^, and 1731 cm^−1^ were derived from the stretching vibration of -OH, -CH, -CH2-, and C=O [31,32,33]. The characteristic peak of AP does not appear in the spectrum of AP-NLC but is highly consistent with the peak position of GMS and the spectrum of blank NLC. This means that AP has been perfectly integrated into the lipid and has not interacted with it. This conclusion can be mutually confirmed by DSC and XRD.

### 2.8. Drug Release Study In Vitro

The in vitro release of AP and AP-NLC was analyzed and compared: There was an abrupt release of free AP, with a cumulative release rate of (62.65 ± 2.07)% in the first 4 h and an average release rate of over 90% in 24 h. In contrast, the cumulative release rate of AP-NLC was lower (Figure 9). Specifically, the cumulative release rates for the first 4h and 24 h were (28.38 ± 0.61)% and (47.73 ± 1.17)%, respectively. The release kinetic model was fitted to the cumulative release curve of AP-NLC, and the results (Table 5) showed that the Ritger–Peppas model had the largest R^2^ value and the best fitting effect. In addition, the “n” value in the fitting equation is 0.27, which is less than 0.45. It is assumed that the AP-NLC diffusion mechanism could be a Fickian diffusion mechanism [34].

### 2.9. Stability of Preparations

Figure 10 is a graphical representation of the changes in particle size, PDI, and zeta potential of AP-NLC when placed at 4 °C for two months. The particle size of freshly prepared AP-NLC was (47.77 ± 1.79) nm, the zeta potential was (−18.5 ± 0.96) mV, and the PDI was (0.217 ± 0.015). The particle size of AP-NLC increased slightly during storage, with the PDI increasing from 0.217 to 0.349, reaching 0.314 on day 50, and the overall uniformity of the distribution decreasing. Zeta potential values did not change significantly, fluctuating in the range of −18.5 to −24.4 mV. The growth in particle size and the increase in the charge carried by the particles may be due to the fusion of the nanoparticles as a result of sedimentation occurring during placement. The above data indicate that the storage stability of AP-NLC is good within the measured time.

### 2.10. Freeze Drying Protection

Freeze drying is a widely employed technique for stabilizing nanoparticles and can help achieve long-term storage of nanoparticles [35]. Additionally, cryoprotectants can be utilized to prevent the aggregation of nanoparticles, ensuring a consistent particle size distribution and average nanoparticle size [36]. Preliminary screening of cryoprotectant types showed that the surface of freeze-dried samples with mannitol added was flat, not shrunken, and not collapsed. When the added concentration was 3%, the lyophilized product was fluffy, full, and delicate (Figure 11).

### 2.11. Safety Evaluation of Preparations

The safety of the preparation was investigated in terms of cell viability and hemolysis assay. The cell viability of NCI-H1299 cells treated with blank NLC was above 80%, which proved that the excipients required for the preparation of the vector were safe and less toxic and excluded the effect of drug excipients on the cells. In Figure 12B, tube 1 was the negative control, with erythrocytes completely aggregated in the lower layer; the upper liquid layer was clear. Tube 2 (positive control) showed a clear red color with complete rupture of erythrocytes. Tubes 3 and 4 were added with AP-NLC and blank NLC, respectively. The hemolysis rate of AP-NLC was (1.34 ± 0.10)%, and that of blank NLC was (2.31 ± 0.22)%, both of which were less than 5%, which indicated that AP-NLC and blank NLC were safe and could fulfill the requirements of injectable drug delivery.

### 2.12. Cell Proliferation Assay

The results of different treatment times showed that the cell viability of each group decreased with the increase in concentration; at the same drug concentration, the prolongation of treatment time resulted in lower viability. Nevertheless, AP-NLC could inhibit the proliferation of NCI-H1299 cells more strongly compared to AP (Figure 13).

### 2.13. Cell Morphology

Figure 14 shows the morphological characteristics of NCI-H1299 cells treated with AP and AP-NLC. Untreated cells were normal in shape, spindle-shaped, and had a strong proliferative ability. After treatment with a low concentration of the drug, the cells lost their original shape, and the tentacles stretched toward polygons. With the increase in drug concentration and culture time, the cells became round, the growth state was poor, the outline between cells became clear, vacuoles and granules appeared in the cytoplasm, and proliferation and adhesion were weakened. When the drug concentration increased to 30 μmol/L, there was a marked change in the number of cells; when it increased to 50 μmol/L, the number of viable cells decreased dramatically, and floating cell debris was everywhere. The above cell morphological changes were more pronounced in the AP-NLC group than in the AP group, demonstrating a stronger cytotoxic effect of AP-NLC, which was consistent with the results of cell viability implemented by CCK-8.

### 2.14. Wound Healing Assay

Figure 15 presents the results of the wound healing assay, demonstrating the migration of NCI-H1299 cells towards the scratched area after treatment with AP and AP-NLC. The scratched area in the control group was almost completely healed at 24 h. The cell migration rates at 12 and 24 h were (12.95 ± 1.80)% and (25.34 ± 2.04)% for the AP group, and (7.15 ± 2.94)% and (11.14 ± 2.66)% for the AP-NLC group, respectively. At 12 h, the difference between the AP-NLC group and the AP group was significant (*p* < 0.05), and the difference between the groups reached a highly significant level (*p* < 0.001) after extending to 24 h. The above results indicated that both AP and AP-NLC could significantly inhibit the migration of NCI-H1299 cells to the scratched area, but the inhibitory effect of AP-NLC was stronger.

### 2.15. Migration and Invasion Assay

Unlike the wound healing assay, which investigates cell migration in the plane, Transwell migration is a simulation of cell migration in 3D. The results of the Transwell migration and invasion can be seen in Figure 16. Compared to the control group, the number of migrating cells was significantly decreased in the AP group, and further reduced in the AP-NLC. AP-NLC markedly inhibited the migratory ability of NCI-H1299 cells, which had been authenticated through the wound healing assay. Subsequently, Matrigel gels were laid in Transwell chambers to simulate cancer cells secreting certain matrix hydrolases in vivo and undergoing infiltration across the epithelial basement membrane. The invasive capacity of the cells was assessed by counting the number of cells penetrating the Matrigel gel in each field of view. The results of the invasion assay were generally consistent with the migration consequences, indicating that AP-NLC could reduce the probability of NCI-H1299 cells undergoing infiltration and decrease the possibility of cell spreading and metastasis.

### 2.16. Plate Cloning formation Assay

The plate cloning formation assay is an effective approach to ascertain the proliferative capacity of individual cells. After continuous treatment of NCI-H1299 cells with low concentrations of AP and AP-NLC, respectively, for 14 days, the clonal communities formed in the AP group were small in size and reduced in number as opposed to untreated cells (Figure 17). However, almost no clonal communities were formed in the AP-NLC group. This exposes the fact that the ability of AP to inhibit the proliferation of individual cells is enhanced when it is encapsulated into NLC.

## 3. Materials and Methods

### 3.1. Materials

Apigenin (AP), glyceryl monostearate (GMS), glyceryl triacetate, poloxamer 188, RPMI 1640 Medium, and Cell Counting Kit-8 (CCK-8) were purchased from Shanghai Titan Scientific Co., Ltd., Shanghai, China. NCI-H1299 cell lines were acquired from Procell Life Science & Technology Co., Ltd., Wuhan, China. Calf serum (CS) was bought from SenBeiJia Biological Technology Co., Ltd., Nanjing, China. SD male rats were purchased from the Mudanjiang Medical University Comparative Medicine Center (Mudanjiang, China).

### 3.2. Preparation of AP-NLC

GMS, glyceryl triacetate, and AP were mixed as the oil phase, which was heated and stirred to melt in a water bath at (85 ± 2) °C. Poloxamer 188 was ultrasonically and uniformly dispersed in ultrapure water as the aqueous phase. Emulsification was initiated by dropwise addition of the aqueous phase at the same temperature to the oil phase under the stirring of a thermostatic magnetic stirrer. It was then sonicated with the JY92-IIDN ultrasonic cell crusher (Shanghai BiLon Instrument Co., Ltd., Shanghai, China), cooled in an ice-water bath, and filtered through a 0.45 μm microporous filter membrane.

A blank nanostructured lipid carrier (blank NLC) was prepared by the same method without adding AP to the oil phase.

### 3.3. Determination of AP content

#### 3.3.1. Choice of Maximum Absorption Wavelength

A series of concentrations of AP methanol solutions were prepared and then scanned at 200–400 nm using a UV-1800 UV-visible spectrophotometer (Shimadzu, Tokyo, Japan), and the UV absorption profiles were recorded.

#### 3.3.2. Drawing of Working Curve

AP was diluted with methanol to a final concentration of 1, 2, 4, 6, 8, 10, and 12 μg/mL and the absorbance was measured at the maximum absorption wavelength. The working curve was plotted, with the AP concentration (μg/mL) as the horizontal coordinate and the absorbance value as the vertical coordinate, and the regression equation was calculated.

#### 3.3.3. Encapsulation Efficiency and Drug Loading of Preparations

AP-NLC was placed in the upper chamber of an ultrafiltration centrifuge tube and centrifuged at 9000 rpm for 10 min, then the liquid in the lower chamber was collected, and the absorbance at 335 nm was measured to calculate the free drug content. The amount of AP added was the same as the total amount of drug. EE% and DL% were computed as follows:(1)$$EE\%=\frac{W_{total}-W_{free}}{W_{total}}\times 100\%$$
(2)$$DL\%=\frac{W_{total}-W_{free}}{W_{total}-W_{free}+W_{lipid}}\times 100\%$$

W_total_: total mass of added AP; W_free_: mass of unencapsulated AP; W_lipid_: total mass of added lipids.

### 3.4. Single-Factor Experiment

With EE% and DL% of AP-NLC as indicators, the ultrasonic power, emulsifier dosage, emulsification time, lipid–drug ratio, and solid–liquid lipid ratio were investigated via a single-factor experiment.

#### 3.4.1. Ultrasonic Power

The drug dosage was fixed at 5 mg, the lipid–drug ratio was 25:1, the solid–liquid lipid ratio was 5:1, the emulsifier dosage was 1%, and the emulsification time was 30 min. The influence of ultrasonic power of 150 W, 300 W, 450 W, and 600 W on EE% and DL% of AP-NLC was investigated.

#### 3.4.2. Emulsifier Dosage

The drug dosage was fixed at 5 mg, the lipid–drug ratio was 25:1, the solid–liquid lipid ratio was 5:1, the emulsification time was 30 min, and the ultrasonic power was 600W. Changes in the amount of emulsifier to 0.5%, 1%, 1.5%, 2%, and 2.5% were investigated, and its influence on EE% and DL% of AP-NLC was observed.

#### 3.4.3. Emulsification Time

The emulsifier dosage was temporarily set at 1%, and other factors remained unchanged. The effects of emulsification times of 10, 20, 30, 40, and 50 min on EE% and DL% of AP-NLC were investigated.

#### 3.4.4. Lipid–Drug Ratio

The emulsification time was set at 20 min, and other factors remained unchanged. The effects of lipid–drug ratios of 20:1, 25:1, 30:1, 35:1, and 40:1 on EE% and DL% of AP-NLC were investigated.

#### 3.4.5. Solid–Liquid Lipid Ratio

The lipid–drug ratio was set at 30:1, and other factors remained unchanged. EE% and DL% of AP-NLC were calculated at solid–liquid lipid ratios of 3:1, 4:1, 5:1, 6:1, 7:1, and 8:1, respectively.

### 3.5. Optimization of AP-NLC Based on CCD-RSM

Based on the single-factor experiment, emulsifier dosage (A), lipid–drug ratio (B), and solid–liquid lipid ratio (C) were finally selected, and these three factors with significant effect on the results were further analyzed in depth. Two response values, EE% (Y_1_) and DL% (Y_2_), were set up and the preparation process was optimized via CCD-RSM. The three-factor, five-level design table is shown in Table 6; the test arrangement and the results are shown in Table 1, Table 2 and Table 3.

### 3.6. Characterization of NLC

#### 3.6.1. Transmission Electron Microscopy

TEM was utilized to observe the morphology of AP-NLC. AP-NLC diluted 100 times with distilled water was dropped on a 400-mesh copper grille, then negatively tinted with 1% phosphotungstic acid. The samples were left to dry before being observed through JEM-2100 Plus transmission electron microscopy (JEOL, Tokyo, Japan).

#### 3.6.2. DSC Analysis

To investigate the thermodynamic characteristics of NLC, about 10 mg of AP, blank NLC, AP-NLC, GMS, and the physical mixture of AP and GMS (1:1, *w*/*w*), respectively, were placed in the crucible. The empty crucible was used as a reference. The above powders were scanned and analyzed by DSC-100L differential scanning calorimetry (Nanjing Dazhan Electrical Technology Company, Nanjing, China) at a heating rate of 15 °C/min, with a scanning range of 30~400 °C.

#### 3.6.3. X-ray Diffraction Study

AP, GMS, the physical mixture of AP and GMS, freeze-dried powder of AP-NLC, and blank NLC were placed on the sample table and lightly compacted. XRD analysis was performed in 2θ mode, scanning from 5° to 90° by XRD-7000 X-ray diffractometer (Shimadzu, Tokyo, Japan).

#### 3.6.4. FT-IR Analysis

AP, GMS, blank NLC, and AP-NLC were respectively crushed with potassium bromide crystals, fully mixed, and then pressed into transparent sheets. The infrared absorption spectrum was analyzed by Nicolet iS5 FT-IR (Thermo Fisher Scientific, Waltham, MA, USA). The wave number scanning range was set to 400–4000 cm^−1^.

### 3.7. Drug Release Study In Vitro

The drug release of AP and AP-NLC was measured in PBS at pH 7.4. Free AP and AP-NLC were put into dialysis bags and released continuously for 24 h at 37 °C, and 3 mL of release medium was removed at 0.5, 1, 2, 4, 6, 8, 10, 12, 16, 20, and 24 h, while equal amounts of isothermal release medium were replenished. The drug release curves were plotted and equations were fitted to the release curves of AP-NLC, respectively. Cumulative drug release rates were calculated and release kinetic models were fitted, including Zero Order, First Order, Higuchi, and Ritger–Peppas.

### 3.8. Stability of Preparations

The stability of AP-NLC was studied via particle size, zeta potential, and the polydispersity index (PDI) by placing samples at 4 °C for two months and measuring every 10 days. Samples were diluted and then particle size, zeta potential, and PDI were determined using a Zetasizer Nano-ZS90 (Malvern Instruments, Malvern, UK).

### 3.9. Freeze Drying Protection

The addition of lyophilized protectants could improve the stability of the preparation. Different types of lyophilization protectants (mannose, trehalose lactose, sucrose, mannitol) were added. The optimal species of lyophilized protectant was first determined, and subsequently, the amount added (3%, 6%, 9%) was explored. The appearance of the lyophilized preparation was considered as an evaluation index.

### 3.10. Cell Culture

NCI-H1299 cells were incubated in a constant temperature incubator at 37 °C containing 5% CO_2_. The medium used was RPMI 1640 containing 10–20% CS and 1% streptomycin and penicillin.

### 3.11. Safety Evaluation of Preparations In Vitro

#### 3.11.1. Cell Viability of Blank NLC

NCI-H1299 cells were inoculated into 96-well plates with 3 × 10^3^ cells per well. A culture medium containing blank NLC of different concentrations (5, 10, 15, 20, 25, 30 μg/mL) was added to the wells. After 24, 48, and 72 h, the drug solution was removed, and then 100 μL of culture medium containing 10% CCK-8 was added. After incubation in the dark for 3 h, the OD value was measured at 450 nm with the Spectra Max M3 microplate reader (Molecular Devices, San Jose, CA, USA). The untreated cells were used as the control, and the culture medium was a blank background. Cell viability was calculated according to the following equation:(3)$$Cell\;Viability(\%)=\frac{{OD}_{test}-{OD}_{blank}}{{OD}_{control}-{OD}_{blank}}\times 100\%$$

#### 3.11.2. Hemolysis Assay

Blank NLC and AP-NLC were incubated with erythrocyte suspensions to assess the safety of hemolysis of the formulations. Blood was collected from SD male rats, placed in an anticoagulant tube, and shaken with glass beads for 5 min to remove fibrin from whole blood. Animal experiments strictly followed the Laboratory Animal Welfare and Ethical Review of Mudanjiang Medical University (authorization number: 20210326-9). Subsequently, the treated blood was centrifuged at 3500 rpm for 5 min, the upper layer of plasma was discarded, and then saline was added and centrifuged after gentle shaking. The above operations were repeated until the supernatant was colorless. The erythrocyte precipitate was diluted with saline to obtain 2% erythrocyte suspension. Different preparations (AP-NLC, blank NLC) were incubated with 2% erythrocyte suspension at 37 °C for 3 h and then centrifuged at 3500 rpm for 5 min. The OD value of the supernatant was measured at 540 nm with the Spectra Max M3 microplate reader. The hemolysis rate is given as below:(4)$$Hemolysis(\%)=\frac{{OD}_{sample}-{OD}_{negative}}{{OD}_{positive}-{OD}_{negative}}\times 100\%$$

### 3.12. Cell Viability Assay

Various concentrations (10, 20, 30, 40, 50, 60 μmol/L) of AP and AP-NLC were co-cultured with NCI-H1299 cells for 24, 48, and 72 h, respectively, and other operations were performed as in Section 3.11.1.

### 3.13. Cell Morphology

NCI-H1299 cells were inoculated in 6-well plates with 1.2 × 10^6^ cells per well. A medium containing different concentrations (0, 10, 20, 30, 40, 50 μmol/L) of AP and AP-NLC, respectively, was added. The morphology of the cells was visualized using the Leica DMI4000B Automatic Inverted Microscope (Leica Microsystems, Wetzlar, Germany) at 0 h, 24 h, 48 h, and 72 h.

### 3.14. Wound Healing Assay

NCI-H1299 cells were seeded in 6-well plates, and when cells fused into a monolayer, equidistant cell scratches were made through a 10 μL pipette tip. Subsequently, the serum-free cell culture medium containing AP and AP-NLC was added separately, and then photographed and recorded at 0 h under an inverted microscope. The wound area was calculated with ImageJ 2.3.0 (National Institutes of Health, Bethesda, MD, USA) [37] and the cell migration rate was calculated with the following formula:(5)$$Cell\;Migration\;Rate(\%)=\frac{{Area}_{0h}-{Area}_{12or24h}}{{Area}_{0h}}\times 100\%$$

### 3.15. Transwell Assays

Distance metastases are very common in lung cancer, accounting for 36.1% of NSCLC patients [38]. The migration and invasion abilities of NCI-H1299 cells were analyzed through Transwell assays. Cells were treated with AP and AP-NLC for 24 h, and trypsinized and suspended in serum-free medium (2 × 10^5^ cells per mL). The cell suspension was added to the Transwell upper chamber, and the lower chamber was placed in a complete medium containing 20% CS and incubated for 24 h. Then, the cells were fixed with 4% paraformaldehyde for 20 min, wiped off the non-migrated cells in the upper chamber, stained with crystalline violet staining solution for 20 min, washed with PBS, and photographed in five randomly selected fields under a microscope. Matrigel matrix was wrapped on a polycarbonate membrane (8 μm pore size) 6 h in advance to carry out the invasion assay; other operations were the same.

### 3.16. Plate Cloning Formation Assay

The plate cloning formation assay is an effective method to determine the proliferation ability of individual cells. After 2000 cells were inoculated in 6-well plates to be plastered, the cells were treated with AP and AP-NLC, respectively. After 14 days of incubation, the cells were fixed with 4% paraformaldehyde and stained with crystalline violet staining solution for 10 min while protected from light. Excess staining solution was washed away, inverted, and dried to take pictures, as well as to count the number of clonal communities.

### 3.17. Statistical Analysis

Data were expressed as mean ± SD (*n* = 3). One-way ANOVA from the GraphPad-Prism 8.0 software (GraphPad Software, La Jolla, CA, USA) was used to calculate the differences between groups. It was statistically significant when *p* < 0.05.

## 4. Conclusions

In this study, AP-NLC was prepared by melt sonication and the prescription was optimized using CCD-RSM. The optimized AP-NLC had a small particle size and suitable PDI and zeta potential, with expected EE% and DL%. AP-NLC can realize a delayed drug release effect, the release model fits well with Ritger–Peppas, and the mechanism of release is Fickian diffusion. Subsequent biological experiments showed that the prepared nanoparticles had better inhibitory effects on proliferation, migration, and invasion of NCI-H1299 cells, which may provide a new possibility for the treatment of NSCLC. In this experiment, the antitumor effects of AP-NLC were investigated only from an in vitro molecular biology perspective, and we will further investigate its in vivo antitumor effects in future research.

## Figures and Tables

**Figure 1 molecules-28-06668-f001:**
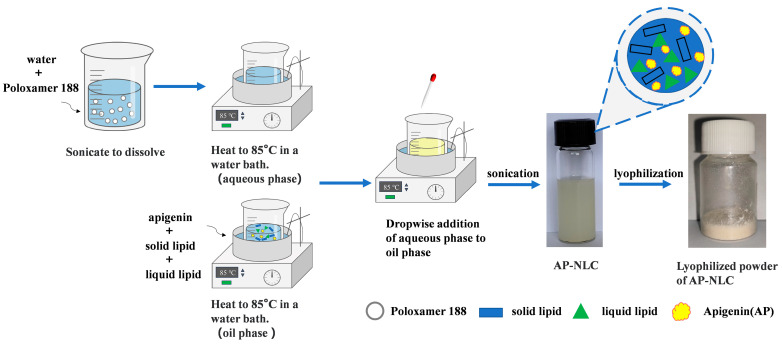
Preparation process of AP-NLC lyophilized powder.

**Figure 2 molecules-28-06668-f002:**
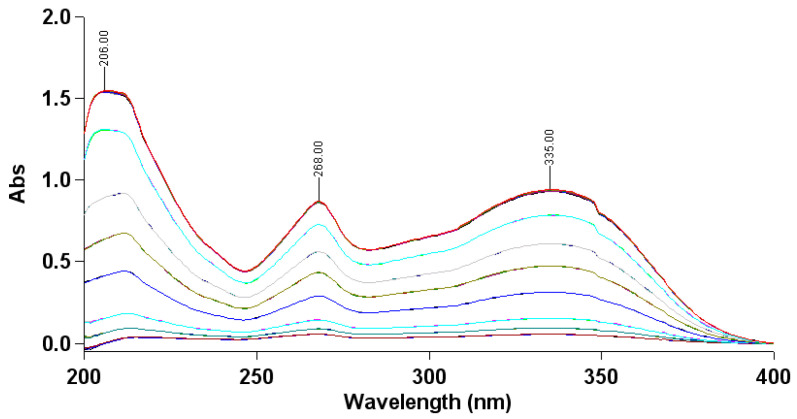
Ultraviolet absorption of the AP–methanol solution within 200–400 nm. (Different color lines represent the absorption curves of different AP–methanol solution concentrations).

**Figure 3 molecules-28-06668-f003:**
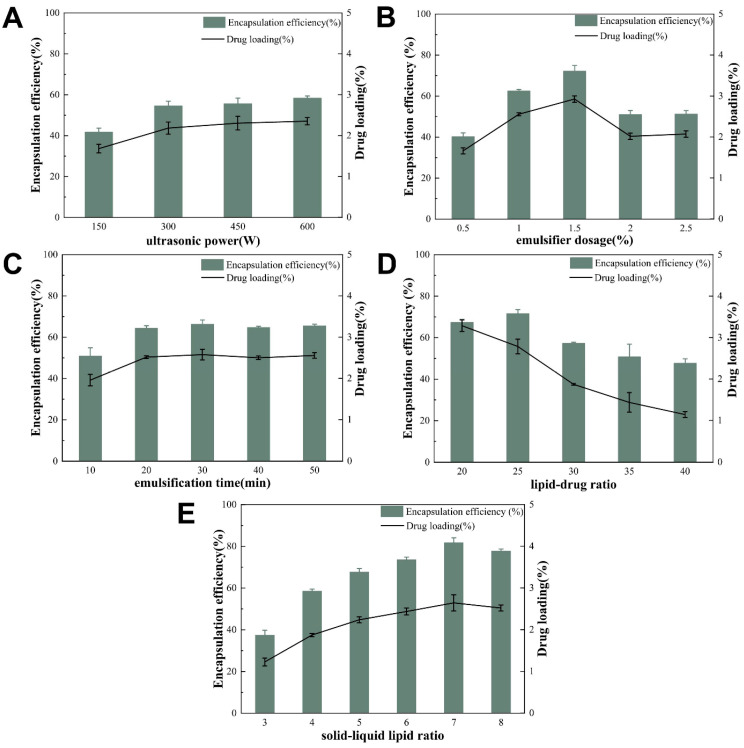
Single-factor results graph. (**A**) ultrasonic power, (**B**) emulsifier dosage, (**C**) emulsification time, (**D**) lipid–drug ratio, and (**E**) solid–liquid lipid ratio.

**Figure 4 molecules-28-06668-f004:**
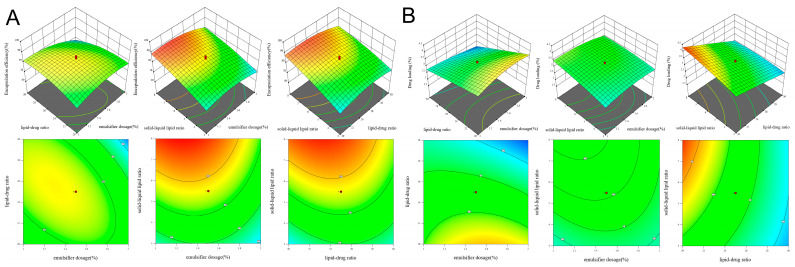
Three-dimensional response surface and contour maps of (**A**) EE% and (**B**) DL% of AP-NLC.

**Figure 5 molecules-28-06668-f005:**
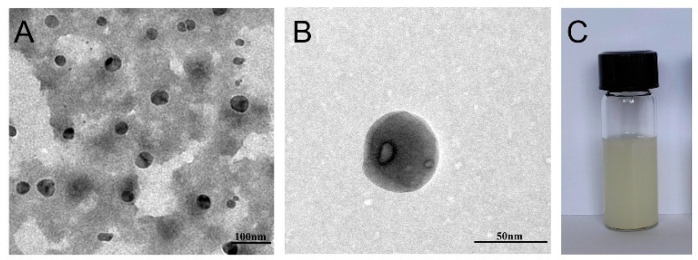
(**A**,**B**) Transmission electron micrograph and (**C**) appearance of AP-NLC.

**Figure 6 molecules-28-06668-f006:**
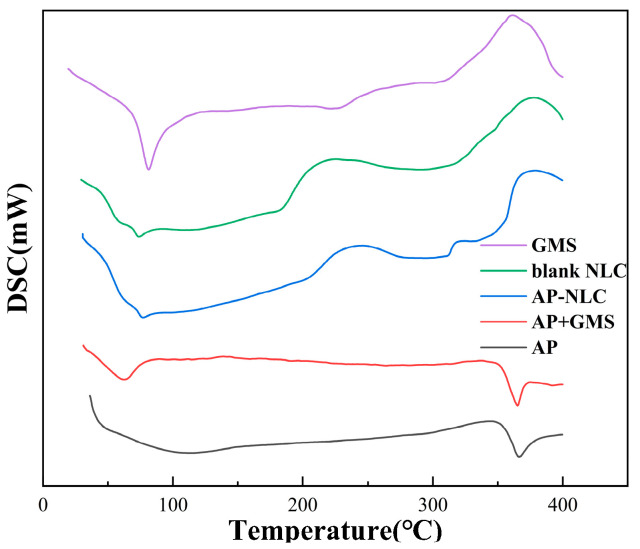
DSC thermograms of AP, AP-NLC, blank NLC, GMS, and the physical mixture of AP and GMS.

**Figure 7 molecules-28-06668-f007:**
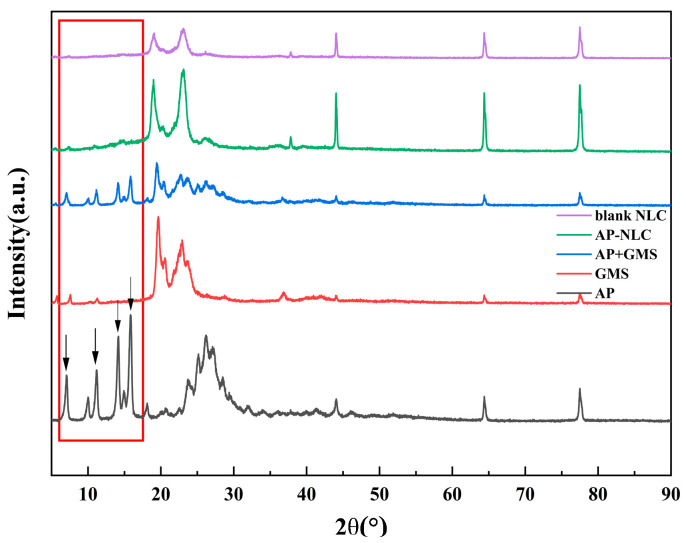
X-ray diffraction pattern of AP, GMS, AP-NLC, blank NLC, and the physical mixture of AP and GMS.

**Figure 8 molecules-28-06668-f008:**
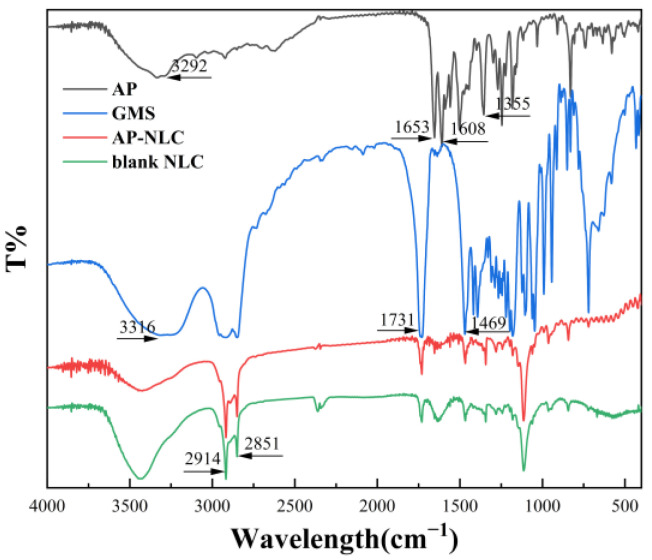
Infrared spectrogram of AP, GMS, blank NLC, and AP-NLC.

**Figure 9 molecules-28-06668-f009:**
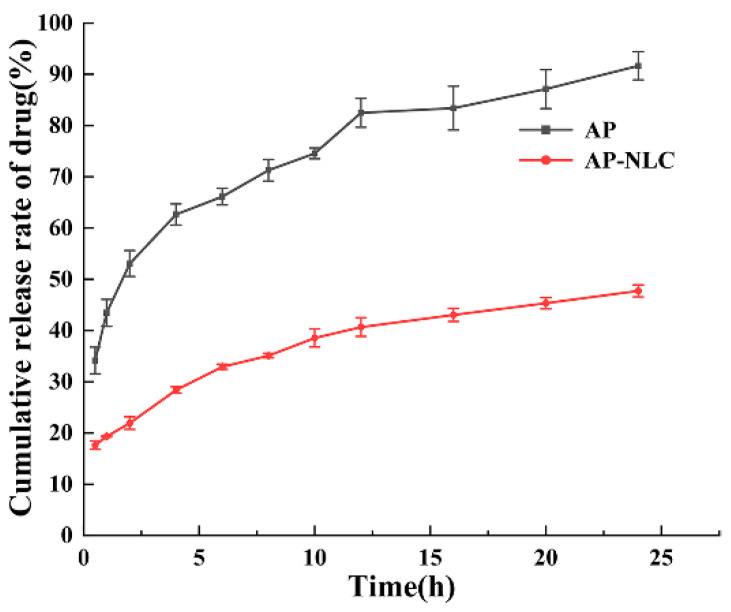
The cumulative release rate of AP and AP-NLC within 24 h under the condition of pH = 7.4.

**Figure 10 molecules-28-06668-f010:**
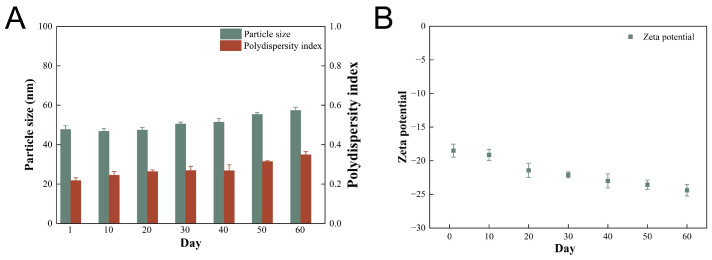
Changes in (**A**) particle size and PDI, and (**B**) zeta potential of AP-NLC over two months.

**Figure 11 molecules-28-06668-f011:**
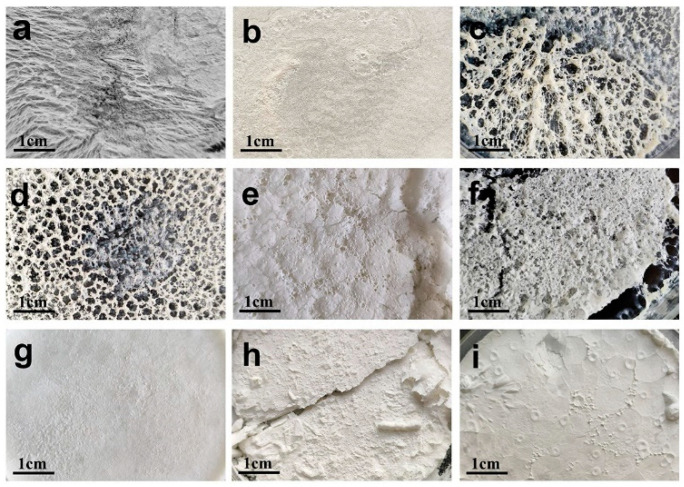
Appearance of lyophilized powder of (**a**) blank NLC, (**b**) AP-NLC, and AP-NLC after the addition of (**c**) 3% mannose, (**d**) 3% trehalose (**e**) 3% lactose, (**f**) 3% sucrose, (**g**) 3% mannitol, (**h**) 6% mannitol, and (**i**) 9% mannitol.

**Figure 12 molecules-28-06668-f012:**
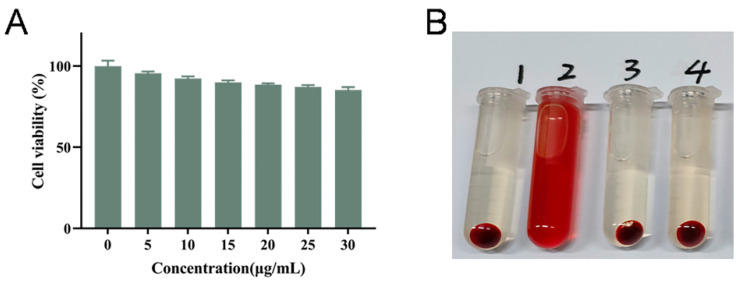
(**A**) Cell survival rate after incubating different concentrations of blank NLC with NCI-H1299 cells. (**B**) A 2% erythrocyte suspension was incubated with (1) physiological saline, (2) distilled water, (3) AP-NLC, and (4) blank NLC.

**Figure 13 molecules-28-06668-f013:**
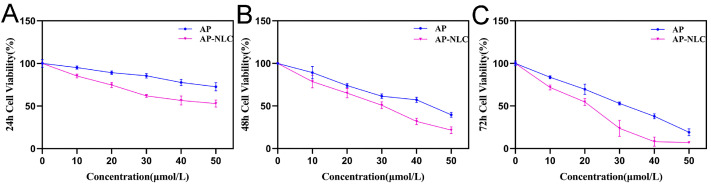
Cell viability after treatment with AP and AP-NLC for (**A**) 24 h, (**B**) 48 h, and (**C**) 72 h.

**Figure 14 molecules-28-06668-f014:**
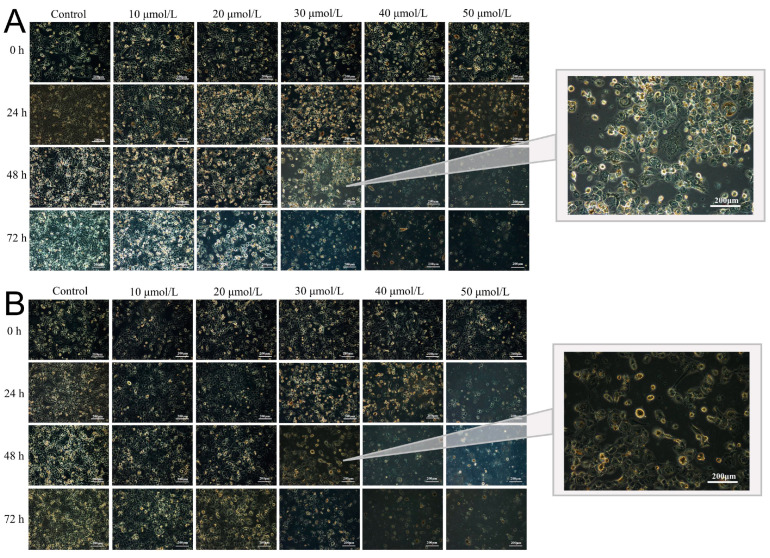
Cell morphology after treatment with different concentrations of (**A**) AP and (**B**) AP-NLC.

**Figure 15 molecules-28-06668-f015:**
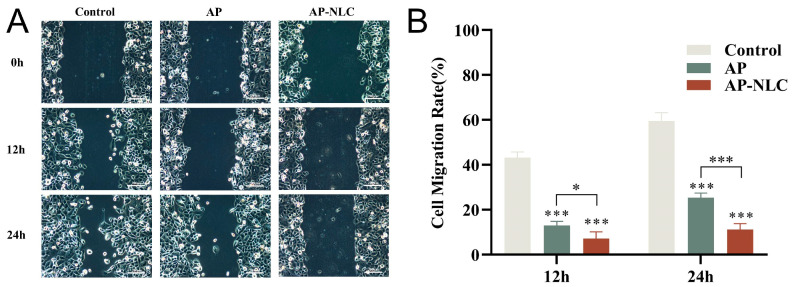
Wound healing assay implemented for NCI-H1299 cells. (**A**) Representative images of cell migration treated with AP and AP-NLC. (**B**) Cell migration rates at 12 and 24 h. Data were expressed as mean ± SD. * *p* < 0.05, *** *p* < 0.001.

**Figure 16 molecules-28-06668-f016:**
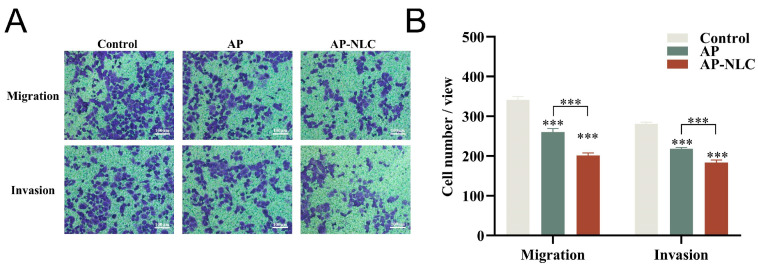
Transwell assay conducted on NCI-H1299 cells. (**A**) Representative images of migration and invasion treated with AP and AP-NLC. (**B**) The average cell number of migration and invasion. Data were expressed as mean ± SD. *** *p* < 0.001.

**Figure 17 molecules-28-06668-f017:**
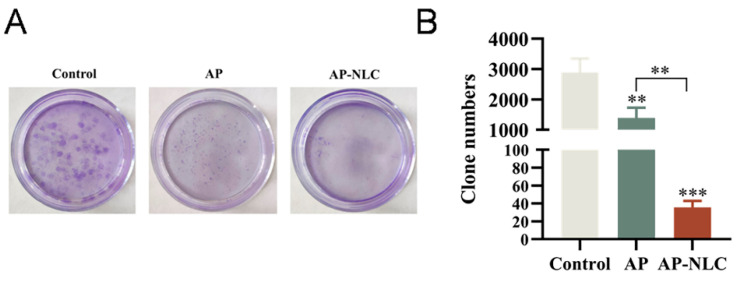
(**A**) The plate cloning formation assay of the control, AP, and AP-NLC groups. (**B**) Clone numbers formed in different treatment groups. Data were expressed as mean ± SD. ** *p* < 0.01, *** *p* < 0.001.

**Table 1 molecules-28-06668-t001:** Test arrangement and response values.

No.	Coded Value			Actual Value			Value of Response	
	A	B	C	A	B	C	Y_1_	Y_2_
1	1	1	−1	2	30	3	64.11 ± 0.33	2.06 ± 0.03
2	0	0	0	1.5	25	5.5	85.59 ± 1.43	3.27 ± 0.03
3	0	0	1.682	1.5	25	9.70	90.10 ± 0.97	3.48 ± 0.04
4	1	−1	−1	2	20	3	67.18 ± 2.40	3.28 ± 0.10
5	−1	1	1	1	30	8	89.27 ± 1.07	2.87 ± 0.14
6	−1.682	0	0	0.66	25	5.5	74.74 ± 1.74	2.89 ± 0.05
7	−1	−1	1	1	20	8	81.82 ± 1.74	3.94 ± 0.07
8	0	−1.682	0	1.5	16.59	5.5	75.75 ± 1.77	4.40 ± 0.13
9	0	0	0	1.5	25	5.5	81.57 ± 0.75	3.28 ± 0.08
10	0	0	0	1.5	25	5.5	83.79 ± 0.64	3.16 ± 0.07
11	0	0	0	1.5	25	5.5	83.14 ± 1.80	3.21 ± 0.06
12	0	0	0	1.5	25	5.5	82.55 ± 1.63	3.19 ± 0.04
13	1.682	0	0	2.34	25	5.5	67.43 ± 2.27	2.63 ± 0.16
14	0	1.682	0	1.5	33.41	5.5	67.41 ± 1.98	2.03 ± 0.12
15	−1	1	−1	1	30	3	74.95 ± 3.17	2.43 ± 0.07
16	−1	−1	−1	1	20	3	63.83 ± 0.96	2.87 ± 0.41
17	1	−1	1	2	20	8	83.68 ± 1.36	3.97 ± 0.10
18	0	0	−1.682	1.5	25	1.30	66.69 ± 0.54	2.60 ± 0.04
19	0	0	0	1.5	25	5.5	83.13 ± 2.16	3.25 ± 0.03
20	1	1	1	2	30	8	69.63 ± 2.25	1.92 ± 0.66

**Table 2 molecules-28-06668-t002:** Variance analysis of Y_1_ regression equation.

Source	Sum of Squares	df	Mean Square	F-Value	*p*-Value	
Model	1380.10	9	153.34	32.50	<0.0001	significant
A	103.32	1	103.32	21.90	0.0009	
B	11.58	1	11.58	2.45	0.1483	
C	642.89	1	642.89	136.26	<0.0001	
AB	159.22	1	159.22	33.75	0.0002	
AC	13.24	1	13.24	2.81	0.1249	
BC	26.83	1	26.83	5.69	0.0383	
A^2^	237.86	1	237.86	50.41	<0.0001	
B^2^	217.81	1	217.81	46.16	<0.0001	
C^2^	31.49	1	31.49	6.67	0.0273	
Residual	47.18	10	4.72			
Lack of Fit	38.09	5	7.62	4.19	0.0710	not significant
Pure Error	9.09	5	1.82			
Cor Total	1427.28	19				
R^2^	0.9669					
Adjusted R^2^	0.9372					
Predicted R^2^	0.7809					

**Table 3 molecules-28-06668-t003:** Variance analysis of Y_2_ regression equation.

Source	Sum of Squares	df	Mean Square	F-Value	*p*-Value	
Model	8.01	10	0.8007	192.45	<0.0001	significant
A	0.1271	1	0.1271	30.54	0.0004	
B	2.81	1	2.81	674.97	<0.0001	
C	0.9176	1	0.9176	220.53	<0.0001	
AB	0.3872	1	0.3872	93.06	<0.0001	
AC	0.1152	1	0.1152	27.69	0.0005	
BC	0.2664	1	0.2664	64.04	<0.0001	
A^2^	0.4701	1	0.4701	112.99	<0.0001	
B^2^	0.0056	1	0.0056	1.35	0.2750	
C^2^	0.0960	1	0.0960	23.08	0.0010	
A^2^B	0.0380	1	0.0380	9.14	0.0144	
Residual	0.0374	9	0.0042			
Lack of Fit	0.0261	4	0.0065	2.88	0.1382	not significant
Pure Error	0.0113	5	0.0023			
Cor Total	8.04	19				
R^2^	0.9953					
Adjusted R^2^	0.9902					
Predicted R^2^	0.9254					

**Table 4 molecules-28-06668-t004:** Validation test results of the AP-NLC prescription.

Indicators	Measured	Predicted	Predicted Error (%)
EE%	88.22 ± 1.61	90.13	2.12
DL%	4.22 ± 0.13	4.40	4.09

**Table 5 molecules-28-06668-t005:** Release fitting model and correlation coefficient of AP-NLC.

Release Kinetic Models	Equation	R^2^
Zero Order	M_t_ = 1.27 × t + 21.75	R^2^ = 0.87972
First Order	Mt = 41.95 × (1-e^−0.37t^)	R^2^ = 0.74094
Higuchi	Mt = 7.53 × t^1/2^ + 12.88	R^2^ = 0.98057
Ritger–Peppas	Mt = 19.59 × t^0.28^	R^2^ = 0.99101

**Table 6 molecules-28-06668-t006:** Factors and levels.

Factor	Name	Level				
		−1.682	−1	0	+1	+1.682
A	emulsifier dosage	0.6591	1	1.5	2	2.34
B	Lipid–drug ratio	16.59	20	25	30	33.41
C	Solid–liquid lipid ratio	1.30	3	5.5	8	9.70

## Data Availability

Not applicable.

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
