# Peer review of "Development of A Nanostructured Lipid Carrier-Based Drug Delivery Strategy for Apigenin: Experimental Design Based on CCD-RSM and Evaluation against NSCLC In Vitro"

_molecules, 2023, doi:10.3390/molecules28186668_

Round 1

Reviewer 1 Report

The manuscript entitled " Development of A Nanostructured Lipid Carrier-based Drug Delivery Strategy for Apigenin: Experimental Design Based on CCD-RSM and Evaluation against NSCLC In Vitro" proposes a new lipid-based nanoparticle system for the delivery of apigenin, exploiting the experimental design to find the optimal formulation.

The authors describe in a rigorous scientific way the experiments conducted to obtain the optimal formulation and outline its characterization.

I take the liberty of indicating only one lack, that relating to the calculation of the percentage yield (Y%) in particles which is not reported in the text, indeed it could be a variable to be included in the experimental design in addition to EE% and DL%.

Although this, I find the paper well written and linear for understanding.

Reviewer 2 Report

In this study, Authors optimized the synthesis conditions of apigenin (AP) loaded nanostructured lipid carrier (AP-NLC) and investigated the anticancer effect of AP-NLC on  NSCLC In Vitro. Basically, the manuscript were well-writen and the obtained data suggested the optimization of AP-NLC synthesis and anticancer effect of AP-NLC was observed against  NCI- 280 H1299 cancer cell lines. However, some major points in the manuscript should be clarified and addressed before it can be considered for publications.

Figure 3 and Table 4 showed that the optimal EE and LD were 90% and 4%, this is quite low, please explain and discuss more about this.

Figure 5: Scale bars should be clearly added

Figure 10A: PDI data was not indicated in Figure 10A, please check and revise. Why the size increased and zeta potential decreased gradually after 2 months, please add discussion.

Figure 11: The lyophilization of nanoparticles solution was investigated to stabilize nanoparticles and can help achieve long-term storage of nanoparticles. However, only the Appearance of lyophilized powder was not enough to support the stability of nanoparticle after lyophilization. The recovery of nano-size after lyophilization should be presented as function of storage time. Scale bars should be added in the figure.

Figure 12 and Figure captions were not fully matched, please check again. In fact the concentrations used to investigate the toxicity (0-30 µg/mL) was quite low, higher concentration should be used. Also, this experiment was to investigate the toxicity, therefore, normal cell lines should be used rather than cancer cells.

Figure 13. Proliferation test should be investigated using normal cells as well. Also the blank NCL sample should be investigated to clearly prove the anticancer activity of AP-NLC

Figure 14: Morphology change should be clearly presented. Higher magnification photos should be shown

Reviewer 3 Report

In general, this is an interesting article showing new methodologies in search for modern carriers of the antitumor drugs. The obtained results confirm objectives.

However, the chemical part is misleading. There are no full chemical names of the compounds used to prepare nanoparticles (NP’s). In particular, triacetin is a triglyceride characterized by high boiling and melting point. Its chemical structure shows that this is an ester of glycerine and acetic but not fatty acid and can not be treated as a lipid. It is commonly used as a relatively inert perfect solvent (e.g. in cosmetica). Since no liquid lipid is used here during NP’s preparation authors should redefine their objectives, change Fig. 1, some Tables and introduce changes in the entire text.

These changes require concept redefinition and essential corrections in the manuscript. I would recommend to reject the paper and encourage authors to resubmit the manuscript again.

Round 2

Reviewer 2 Report

Authors have revised the manuscript and clarified my comments. I have no further comments on revised manuscript

Author Response

Thank you very much for your comments and professional advice. These opinions help improve academic rigor of our article. We will take your previous comments and apply them to subsequent experiments.

Reviewer 3 Report

Chemical text books and also for example Encyclopedia Brytannica give definition of a group of lipids belonging to a triacylglycerides as esters of glycerine and fatty acids. Essential there is further definition of fatty acids. Until ca 10 years ago these were linear carboxylic acids with at least 5-7 carbon atoms in the side chain. Than in some publications of the nutrition science and field new term was introduced: short side chain fatty acids, including forming acid with no side chain et al. That was probably moment that was overlooked by chemists to clarify the definition. Particularly, that one of the  critical characteristics of fatty acids is lack of solubility in water, what is not the case for formic, acetic, propionic, etc acids.

Anyway, it is nowadays relatively easy to introduce new term in certain area, not reviewed by competent reviewers from another field.

Reviewer was aware of the common classification of Triacetin as lipid, but was hoping at least for some reflection.

Few other points can be addressed in the new version of this manuscript:

1. P.2/69 and 70: “...the maximum EE% and DL%, which is of extraordinary significance in maximizing …” and “...DoE can be implemented by factorial design...”

Give definitions for EE%, DL% and DoE.

2. P. 2/71: “...CCD-RSM.…”

Give full description and then abbreviation.

Summing up, minor revision will enhance paper readability.

Author Response

First of all, we gratefully thank you for the time spend making your constructive remarks and useful suggestions, which has significantly raised the quality of the manuscript. Secondly, we appreciate your understanding of the definition for compounds and your serious and rigorous attitude towards research.

Definitions for EE%, DL%, and DoE have been added. Changes have been made based on your comments and you can review them in review mode.